# Conversion of Retinyl Palmitate to Retinol by Wheat Bran Endogenous Lipase Reduces Vitamin A Stability

**DOI:** 10.3390/foods13010080

**Published:** 2023-12-25

**Authors:** Eline Van Wayenbergh, Jonas Blockx, Niels A. Langenaeken, Imogen Foubert, Christophe M. Courtin

**Affiliations:** 1Laboratory of Food Chemistry and Biochemistry & Leuven Food Science and Nutrition Research Centre (LFoRCe), Department of Microbial and Molecular Systems (M2S), KU Leuven, Kasteelpark Arenberg 20, B-3001 Leuven, Belgium; eline.vanwayenbergh@kuleuven.be (E.V.W.); niels.langenaeken@kuleuven.be (N.A.L.); 2Research Unit of Food and Lipids & Leuven Food Science and Nutrition Research Centre (LFoRCe), Department of Microbial and Molecular Systems (M2S), KU Leuven KULAK, Etienne Sabbelaan 53, B-8500 Kortrijk, Belgium; jonas.blockx@kuleuven.be (J.B.); imogen.foubert@kuleuven.be (I.F.)

**Keywords:** wheat bran, vitamin A, stabilisation, storage, lipase, lipid oxidation

## Abstract

Wheat bran can be used as a cost-effective food ingredient to stabilise vitamin A. However, wheat bran endogenous enzymes have been shown to reduce vitamin A stability. In this study, we elucidated the mechanism for this negative effect in an accelerated storage experiment with model systems consisting of native or toasted wheat bran, soy oil and retinyl palmitate (RP). Both native and toasted wheat bran substantially stabilised RP. While RP was entirely degraded after ten days of storage in the absence of wheat bran, the RP retention after ten days was 22 ± 2% and 75 ± 5% in the presence of native and toasted bran, respectively. The significantly stronger stabilising effect of toasted bran was attributed to the absence of bran endogenous enzymes. In contrast to toasted bran systems, noticeable free fatty acid production was observed for native bran systems. However, this did not result in a pronounced lipid oxidation. Next to lipid hydrolysis, wheat bran lipase was shown to hydrolyse retinyl esters to the less stable retinol and fatty acids. This reaction could explain the major part, about 66 ± 5%, of the difference in RP stabilisation between native and toasted wheat bran.

## 1. Introduction

Vitamin A deficiency is a major public health problem. Several effective strategies to combat vitamin A deficiency have already been established. These include the enrichment of food products with vitamin A and the regular consumption of vitamin A supplements [1,2,3]. Retinyl esters, including retinyl palmitate (RP), are mostly used to produce vitamin A supplements and fortified food products. These esterified vitamin A forms possess higher stability compared to the free form, retinol [4]. However, the stability of RP is still relatively low as its chemical structure contains a high number of unsaturated bounds, which renders RP susceptible to oxidative degradation. This limits the effectiveness of food fortification [5].

Different methods for vitamin A stabilisation have already been developed. These include the incorporation of antioxidants and the use of a wide range of nano-based delivery systems [5,6]. However, advanced techniques, like nano-based delivery systems, are often costly, and information regarding their performance in food systems is rather limited, which hampers their application in the food industry [7]. We have shown in a previous study that cereal bran can serve as a natural, healthy and cost-effective stabiliser for preserving vitamin A during storage [8]. However, we observed that the presence of wheat bran endogenous enzymes limits this stabilising effect [9]. This negative impact of enzymatic activity on vitamin A stability is not yet understood.

We here hypothesise that two mechanisms are at play. The first, indirect mechanism comprises the acceleration of lipid oxidation by bran endogenous enzymes. This presumably leads to accelerated vitamin A degradation as more extensive lipid oxidation has already been shown to result in faster vitamin A degradation [10]. We hypothesise that three wheat bran endogenous enzyme types, lipases, lipoxygenases and phytases, play an important role in promoting lipid oxidation. Firstly, lipases are thought to promote lipid oxidation by the production of free fatty acids (FFAs), which are more susceptible to oxidation than triacylglycerols (TAGs) [11]. However, it should be noted that studies investigating the pro-oxidative effect of FFAs show inconsistent results [12,13,14]. Secondly, lipoxygenases promote enzymatic lipid oxidation and can cause the co-oxidation of vitamin A [15]. Lastly, phytases degrade phytic acid, which results in the release of free, pro-oxidative minerals that can promote lipid and vitamin A oxidation [16]. The second, direct pathway comprises the lipase-catalysed conversion of RP to palmitic acid and retinol. As the stability of retinol is lower than that of RP [4], this conversion would lead to faster vitamin A degradation. It has been shown by Turner et al. [17] that some lipases can hydrolyse the ester bond of RP. However, it is not known if wheat bran lipase, in particular, is able to hydrolyse this ester bond.

Against this background, the goal of this study is to gain in-depth insights into the relationship between the wheat bran endogenous enzymatic activity and RP stability during storage. To this end, a toasting process is used to inactivate wheat bran endogenous enzymes, and toasted wheat bran is compared to native wheat bran. Although toasting might affect other wheat bran properties such as the antioxidant capacity, we can state, based on our previous studies, that enzyme inactivation is the main effect of toasting influencing vitamin A stability [9,18]. This study focuses on the effect of wheat bran lipase by monitoring the lipase-catalysed reactions, shown in Figure 1, during storage. The activity of other wheat bran endogenous enzymes is not investigated in depth. The goal of this study can be divided into two sub-objectives. The first objective is to investigate the effect of wheat bran lipases on lipids and the relationship between lipid oxidation and RP degradation. Hereto, the RP degradation, FFA production and lipid oxidation are monitored during the accelerated storage of systems composed of RP, soy oil and native or toasted wheat bran. In line with the indirect mechanism mentioned above, we hypothesise that a native wheat bran system will show pronounced free fatty acid production, together with more substantial lipid oxidation and faster RP degradation compared to a toasted wheat bran system. The second objective is to investigate the direct impact of lipase on RP by studying the conversion of in-house-produced retinyl tridecanoate (RTD) to retinol and tridecanoic acid during storage. RTD is used instead of RP, as tridecanoic acid is not inherently present in wheat bran lipids or soy oil. Palmitic acid, on the other hand, is present in soy oil and wheat bran lipids [19,20], making it almost impossible to distinguish between free palmitic acid originating from lipid hydrolysis and free palmitic acid originating from retinyl ester hydrolysis. In line with the direct mechanism mentioned above, we hypothesise that wheat bran lipase can hydrolyse the ester bond of RP or RTD. This conversion presumably results in a faster breakdown of vitamin A for native compared to toasted wheat bran systems.

The insights gained in this study will shed light on the effect of enzymatic activity on RP stability. This information is crucial to improving RP stability, not only in the investigated model systems but also in various RP-fortified food products where enzymes are active during food processing.

## 2. Materials and Methods

### 2.1. Materials

All-*trans*-retinol (RE) (≥95% purity), retinyl acetate (99.7% purity), retinyl palmitate (RP) (≥93% purity), N,N-di-isopropylethylamine (DIPEA, 99.5% purity) and tridecanoic acid (≥99% purity) were supplied by Merck KGaA (Darmstadt, Germany). Soy oil was obtained from Vandemoortele (Izegem, Belgium) and commercial wheat (*Triticum aestivum* L.) bran was from Dossche Mills (Deinze, Belgium). The approximate composition of wheat bran, as supplied by the supplier, and the fatty acid profile of soy oil, measured as described in Section 2.9, can be found in Appendix A, respectively. Impurities in soy oil, like surfactants, were removed using the procedure of Bahtz et al. [21] with some small adaptations. In short, 10 g Florisil^®^ was added to 50 mL soy oil. After shaking for 2 h, the oil was filtered and kept at −80 °C. Cholesterol, trilinolein and linoleic acid (C18:2) were supplied by Larodan (Solna, Sweden), lauric acid (C12:0) was from Nu-chek Prep. Inc. (Elysian, MN, USA) and tridecanoyl chloride (≥99% purity) was from Santa Cruz Biotechnology (Heidelberg, Germany). Acetic acid, dichloromethane, chloroform, hexane, methanol and sulfuric acid (≥95%) were supplied by Acros Organics (Geel, Belgium). Toluene was from Chem-Lab nv (Zedelgem, Belgium) and propan-2-ol was from Honeywell Riedel-de Haën (Seelze, Germany). All other chemicals, solvents and reagents were from Merck KGaA (Darmstadt, Germany). The solvents used were HPLC-grade.

### 2.2. Toasting of Wheat Bran

Wheat bran was toasted to inactivate endogenous enzymes. To this end, a fine layer of wheat bran was applied onto sieves (mesh size: 38 μm) and subjected to toasting in an oven (Memmert GmbH, Schwabach, Germany) at 170 °C for 30 min. This toasting process resulted in the inactivation of peroxidase, the most thermostable wheat bran endogenous enzyme [22]. Peroxidase inactivation was verified according to Bergmeyer [23]. For native wheat bran, a peroxidase activity of 0.97 ± 2.56 units per gram was measured, whereas no peroxidase activity was measured after toasting (0.00 ± 0.1 units per gram).

### 2.3. Synthesis of Retinyl Tridecanoate

Retinyl tridecanoate (RTD) was synthesised in-house based on a method by Austin-Brown and Chapman [24] with some notable adjustments. Tridecanoyl chloride (0.15 mL, 5.89 × 10^−4^ mol) was dissolved in 100 mL dichloromethane in a 250 mL round-bottom flask. Retinol (144.2 mg, 5.03 × 10^−4^ mol) was added to the mixture, followed by DIPEA (0.1 mL, 5.74 × 10^−4^ mol). The mixture was gently swirled around in the flask for 15 min at room temperature (20 °C). The solution was washed three times with 100 mL Milli-Q water, using a separation funnel, after which the solution was dried over sodium sulphate. Next, the solvent was removed using a rotary evaporator (40 °C, 50 mbar). A dark yellow, clear, viscous liquid was obtained with a net mass of 275.5 mg. The purity of the retinyl tridecanoate is estimated at 85 ± 5%. Additional information on the characterisation of the in-house synthesised RTD can be found in the Appendix A.

### 2.4. Accelerated Storage Experiments

The first accelerated storage experiment was performed with three samples, a control sample of RP-enriched soy oil (0.80% RP), a system composed of RP (0.16%), soy oil (19.84%) and native wheat bran (80.00%), and a system composed of RP (0.16%), soy oil (19.84%) and toasted wheat bran (80.00%) (Samples 1, 2 and 3, Table 1). The composition of the systems used in this study, in terms of concentration RP, oil and wheat bran, was chosen based on our previous study [8]. To prepare these systems, 0.3 g RP was dissolved in 37.5 g purified soy oil. Next, RP-enriched oil (1.0 g) was mixed with native or toasted wheat bran (4.0 g) using a 10 g pin mixer (National Manufacturing, Lincoln, NE, USA). For each system, two mixtures of 5 g were made, pooled and split into 10 portions of 1 g each. The RP-enriched oil was divided into 10 portions of 3 g each and served as a control sample. For each sample, one portion was frozen at −80 °C. This sample corresponded to 0 days of storage (starting point). The other nine portions were stored in the absence of light, in open, plastic containers (h: 6.8 cm, d: 3.0 cm) in a climate chamber (Memmert GmbH, Schwabach, Germany) at 60 °C and 70% relative humidity for 1, 3, 7, 10, 14, 21, 28, 42 and 56 days. The samples were stored at an elevated temperature and high relative humidity to accelerate chemical degradation reactions, such as oxidation. After storage, the samples were stored at −80 °C prior to further analysis. The RP content, FFA content, TAG content, peroxide value and ratio of unsaturated over saturated fatty acids were followed up during storage according to the procedures described below (Section 2.5, Section 2.6, Section 2.7, Section 2.8 and Section 2.9).

The second, follow-up storage experiment was performed using eight samples, of which the composition is given in Table 1. Hereby, the molar concentrations of RP, RE and RTD in soy oil were kept constant. The samples were prepared as described above, divided into 10 portions of 1 g each for the samples with bran and 1.5 g each for the samples without bran, and stored in the absence of light in open, plastic containers at 60 °C and 70% relative humidity for 0, 6, 12 and 24 h, and 3, 5 and 7 days. The RE and RP contents were followed up during storage for samples 1 to 6 as described in Section 2.5. The free tridecanoic acid content was monitored during storage for samples 7 and 8, as described in Section 2.10.

### 2.5. Quantification of Retinol (RE) and Retinyl Palmitate (RP)

The RE and RP in the stored samples and the samples that were not stored were quantified as described in Van Wayenbergh et al. [25]. This procedure, originally developed for the analysis of RP, was extrapolated to RE by preparing a calibration curve for RE and checking the RE recovery. No modification of the extraction procedure or HPLC method was needed to analyse RE. In short, RP or RE was extracted from a 100.0 mg sample using two consecutive extractions, and retinyl acetate was added as the internal standard. In each extraction step, 5.0 mL acetone/methanol (7/3, volume-to-volume (*v*/*v*)) was added and the samples were vortexed for 30 s and centrifuged (750× *g*, 10 min, 20 °C). The supernatant was decanted and both extracts were pooled. For the samples consisting of RE- or RP-enriched oil, 25.0 mg oil was dissolved in 5.0 mL acetone/methanol (7/3, *v*/*v*). RE and RP were quantified using reversed-phase high-performance liquid chromatography (HPLC) with UV detection (325 nm) (Shimadzu, Kyoto, Japan). To this end, a Kinetex C18 column (5 μm, 100 Å, 150 × 4.6 mm, Phenomenex, Torrance, CA, USA) was used and a tertiary gradient elution was applied. The elution solvents were methanol, Milli-Q water and methanol/methyl tert-butyl ether (50/50, *v*/*v*), the injection volume was 10 μL, and the flow rate was set at 1 mL/min. After the determination of the RE and RP contents, the percentage of RE or RP retention was determined as:$$RE\;or\;RP\;retention\;\left(\%\right)=\frac{{\left[RE\;or\;RP\right]}_{at\;time\;t}}{{\left[RE\;or\;RP\right]}_{at\;time\;0}}\times 100$$

[RE] and [RP] are the measured RE and RP concentrations at the respective time points, expressed in μg/g.

### 2.6. Lipid Extraction

Lipids were extracted from a 50.0 mg sample preceding the analysis of the FFA and TAG contents (Section 2.7), the peroxide value (Section 2.8), the ratio of unsaturated over saturated fatty acids (Section 2.9) and the free tridecanoic acid content (Section 2.10). To this end, the chloroform/methanol extraction procedure described in Ryckebosch et al. [26] was used, with some small adaptations as described in Van Wayenbergh et al. [18].

### 2.7. Quantification of Free Fatty Acids and Triacylglycerols

The FFA and TAG contents of the extracted lipids (Section 2.6) were analysed by HPLC with evaporative light scattering detection (ELSD). The FFA production over storage time serves as a measure for wheat bran endogenous lipase activity. Calibration curves were made with linoleic acid (C18:2) and trilinolein to quantify FFAs and TAGs, respectively. The lipids were dissolved in chloroform. Part of this lipid solution, which contained approximately 2.0 mg extracted lipids, was brought into an amber glass vial. Chloroform was evaporated using the Rotational Vacuum Concentrator (2 mbar, 40 °C), the exact mass of lipids in the vial was determined gravimetrically and 40 μL cholesterol in chloroform (50 mg/mL) was added as the internal standard. After the evaporation of chloroform, the residue was redissolved in 1.0 mL isooctane and FFAs and TAGs were quantified by HPLC-ELSD as described in Melis et al. [27]. Hereto, a quaternary gradient with isooctane, acetone/ethyl acetate (2/1, *v*/*v*) containing 70 mM acetic acid, 2-propanol/water (85/15, *v*/*v*) containing 7.5 mM acetic acid and 7.5 mM triethylamine, and 2-propanol was used. The HPLC system (Shimadzu, Kyoto, Japan) was equipped with an Alltech Model 3300 ELSD (Büchi, Hendrik-Ido-Ambacht, The Netherlands). A polar monolithic Chromolith Performance–Si column (100 mm × 4.6 mm) (Merck KGaA, Darmstadt, Germany) was used and the injection volume was 2 μL. The FFA and TAG contents were expressed as mg per 100 mg lipids.

### 2.8. Primary Oxidation: Peroxide Value

The ferrous oxidation-xylenol orange (FOX) method described in Gheysen et al. [28] was used to monitor the peroxide value during storage. Briefly, 4.95 mL chloroform/methanol (7/3, *v*/*v*) was used to dissolve 5.0 mg of extracted lipids (Section 2.6) or RP-enriched soy oil. This solution was diluted by 1/2 for bran-containing samples and 1/45 for the control sample consisting of RP-enriched soy oil to obtain a 9.9 mL diluted lipid solution. The absorbance of this diluted lipid solution was measured at 560 nm. Xylenol orange (50 μL, 10 mM) and Fe^2+^ chloride solution (50 μL,18 mM, acidified with 1 μL 10 mM HCl) were added. After exactly 5 min incubation at room temperature, the absorbance was measured at 560 nm. The measured absorbance was corrected for the absorbance of the diluted lipid solution and the Fe^2+^ solution blank. Next, the peroxide value was calculated based on a calibration curve set up with an Fe^3+^ standard solution and expressed in micro-equivalents (μeq) hydroperoxides per g lipids. Delta values were calculated to express the measured peroxide values relative to the peroxide value at the starting point.

### 2.9. Determination of the Ratio of Unsaturated over Saturated Fatty Acids

The fatty acid composition of the extracted lipids or oil was determined by performing acid-catalysed methylation, whereby fatty acids, mainly present in TAGs, are released and methylated. The formed fatty acid methyl esters (FAMEs) were then analysed by gas chromatography as described in Gheysen et al. [29]. The extracted lipids here consist of wheat bran lipids, soy oil lipids and RP, whereas the oil consists of soy oil lipids and RP. Lauric acid (C12:0) was used as an internal standard, and the peaks were identified using a set of FAME standards (Nu-Chek Prep. Inc., Elysian, MN, USA). The fatty acid content was expressed as g/100 g lipids. Based on these results, the ratio of unsaturated over saturated fatty acids was determined. This ratio was used to complement the measured peroxide values (Section 2.8), making the measurement of secondary oxidation products unnecessary for this study. Unsaturated fatty acids are susceptible to oxidation, but saturated fatty acids are not. Consequently, extensive lipid oxidation will decrease the ratio of unsaturated over saturated fatty acids over storage time.

### 2.10. Quantification of Free Tridecanoic Acid

The free tridecanoic acid content of the lipids extracted from samples 7 and 8 (Table 1), as described in Section 2.6, was determined by liquid chromatography mass spectrometry (LC-MS). To this end, lauric acid (C12:0) was used as an internal standard and was added to each sample or calibration solution at a concentration of 5 μg/mL. A calibration curve of tridecanoic acid with levels 250 μg/mL, 50 μg/mL, 5 μg/mL, 1 μg/mL, 0.5 μg/mL and 0.05 μg/mL was made in toluene/methanol (1/2, *v*/*v*). The extracted lipids were diluted to a concentration of exactly 495 μg/mL in toluene/methanol (1/2, *v*/*v*) to a total volume of 1.5 mL.

The method for the quantification of tridecanoic acid was based on the method of Van Meulebroek et al. [30], which was substantially adapted and optimised. This optimised method is briefly described below. Compound separation was achieved by UHPLC on a Vanquish Flex UHPLC system (Thermo Fischer Scientific Waltham, MA, USA), with a binary pump system for the mixture of two solvents: 20 mM ammonium acetate in Milli-Q water (solvent A) and 20 mM ammonium acetate in methanol (solvent B). A gradient was applied (expressed in % of solvent B): 1′/75; 2′/90; 6′/98; 15′/100; 22′/100; 22′/75, for a total run time of 25 min. The solvent flow was put at 0.3 mL/min. An Accucore C18 reversed-phase column (2.6 μm, 80 Å, 2.1 × 100 mm) fitted with a universal Uniguard column (Thermo Fischer Scientific, holder for 2.1/3.0 MM) was used and operated at 40 °C. The injection volume was set at 5 μL. The calibration solutions were injected in duplicate. Analysis of the different separated components was performed on a Q Exactive MS system with electrospray ionisation (ESI) and an orbitrap detector. The MS was run in negative ion mode for the optimal detection of FFAs, as ionisation happens due to the loss of a proton. The used scan range was 153.4–2300 *m*/*z* with a sheath gas flow rate of 60 arbitrary units (au), auxiliary gas flow rate of 20 au and sweep gas flow rate of 2 au. The capillary temperature was set at 285 °C while the auxiliary gas heater temperature was set at 370 °C. The mass resolution was 70,000 (Hz) full width half maximum. The ESI spray voltage was set at 3 kV and the S-Lens radio frequency level was put at 70 to enhance sensitivity. With these settings, fragmentation was avoided as much as possible.

Data analysis was performed using TraceFinder 5.1 software. The analysed components, lauric acid (C_12_H_24_O_2_; M-H^+^) and tridecanoic acid (C_13_H_26_O_2_; M-H^+^), had *m*/*z* values of 199.170 and 213.186, respectively, with an allowed variation of 3 ppm. The peak areas of both components were integrated and used for further analysis. Correction for technique variation was done by dividing the tridecanoic acid peak areas by the corresponding lauric acid peak area to obtain response factors (RF). A linear calibration curve was subsequently created based on the RFs of the six calibration levels. Quantitative determination of tridecanoic acid was achieved using the calibration curves. Both the measured tridecanoic acid content and the initially present RTD content were converted from mg/g lipids to moles/g lipids, and corrected for the measured free tridecanoic acid content present at the start of the storage experiment. The percentage of RTD hydrolysis was calculated as the ratio of the measured tridecanoic acid content (moles/g lipids) over the initially present RTD content (moles/g lipids).
$$RTD\;hydrolysis\;\left(\%\right)=\frac{{Free\;tridecanoic\;acid\;content}_{at\;time\;t}}{{RTD\;content}_{at\;time\;0}}\times 100$$

### 2.11. Statistical Analysis

The storage experiments were performed in duplicate. Analytical measurements on each sample were also performed in duplicate, resulting in a 2 × 2 setup. Standard deviations, therefore, represent standard deviations of duplicate storage experiments.

## 3. Results and Discussion

### 3.1. Vitamin A Degradation during Accelerated Storage

The RP contents of three different samples, RP-enriched soy oil, RP-enriched soy oil mixed with native wheat bran and RP-enriched soy oil mixed with toasted wheat bran, were monitored during the first accelerated storage experiment (Figure 2). As shown in Figure 2, RP was fully degraded after 10 days in the control sample, which did not contain wheat bran. In the presence of native wheat bran, the RP retention after 10 days of storage was 22 ± 2%, showing the stabilising effect of wheat bran on vitamin A. In the presence of toasted wheat bran, the stabilising effect was much more pronounced, as 75 ± 5% RP was retained after 10 days of storage. After eight weeks of storage, the RP retention in the toasted wheat bran sample was 29 ± 2%, while RP was fully degraded after four weeks of storage in the native wheat bran sample.

We hypothesised that the large difference in RP stabilisation between native and toasted wheat bran can be explained by the difference in wheat bran endogenous enzymatic activity, in particular wheat bran endogenous lipase activity. Two different mechanisms for the negative effect of lipase on RP stability were suggested. The first mechanism involves the acceleration of lipid oxidation by the release of free fatty acids resulting from lipase-catalysed lipid hydrolysis, and the subsequent acceleration of RP oxidation caused by the high abundance of lipid oxidation products. In addition, wheat bran endogenous lipoxygenases and phytases are also thought to contribute to the acceleration of lipids and RP. However, these enzymes are not investigated in depth in this study. The second mechanism involves the direct action of lipase on RP, resulting in the hydrolysis of RP to the more labile molecule retinol and palmitic acid. These two possible mechanisms will be discussed in Section 3.2 and Section 3.3, respectively.

### 3.2. The Relationship between Free Fatty Acid Production, Lipid Oxidation and Vitamin A Degradation

The FFA content, TAG content, peroxide value and fatty acid profile were monitored during storage for the three types of samples from the first storage experiment. The results for the FFA content and TAG content are shown in Figure 3, and the results for the peroxide value and the ratio of unsaturated over saturated fatty acids are shown in Figure 4.

As shown in Figure 3, the FFA content of the sample with native wheat bran increased from 2.1 ± 0.2 mg/100 mg lipids before storage to 47 ± 14 mg/100 mg lipids after eight weeks of storage. Meanwhile, the TAG content decreased from 74 ± 8 mg/100 mg lipids before storage to 9.8 ± 0.6 mg/100 mg lipids after eight weeks of storage. This indicates that TAGs in soy oil and/or wheat bran are hydrolysed into FFAs and glycerol by wheat bran lipase under the storage conditions used. No increase in FFA content was observed over time for the samples without wheat bran and with toasted wheat bran, which was expected due to the absence of lipase activity. However, the sample without bran showed a strong decrease in TAG content over time, from 65 ± 3 mg/100 mg lipids before storage to 3.5 ± 0.3 mg/100 mg lipids after eight weeks. The toasted wheat bran sample showed a less pronounced decrease in TAG content from 70 ± 8 mg/100 mg lipids to 54.0 ± 0.8 mg/100 mg lipids after eight weeks of storage. The decrease in TAG content for both samples could not be explained by the hydrolysis of TAGs to FFAs and glycerol. Presumably, the decrease in TAG content was caused by the oxidation of TAGs. To verify this, the peroxide value, a measure for primary lipid oxidation, was monitored as a function of storage time.

For the control sample, the decrease in TAG content was accompanied by extensive lipid oxidation, as measured by a substantial increase in peroxide value up to 1244 ± 245 μeq hydroperoxides/g lipids after two weeks (Figure 4A). The extensive lipid oxidation was also reflected in the fatty acid profile (Figure 4C), as the ratio of the unsaturated fatty acids over the saturated fatty acids decreased from 6.0 ± 0.2 to 4.61 ± 0.02 after two weeks of storage and to 2.10 ± 0.02 after four weeks of storage. This decrease, together with the strong increase in peroxide value, indicates that extensive lipid oxidation had occurred. It has been shown in the literature that extensive lipid oxidation is associated with rapid vitamin A degradation in fortified oils [10,31]. This relationship is in line with the results presented here. It is hypothesised that intermediates of the lipid oxidation pathway interact with RP and subsequently cause RP degradation.

Although a clear increase in FFAs was noticed for the sample with native wheat bran, the degree of lipid oxidation measured for this sample was very limited and similar to the sample with toasted wheat bran (Figure 3A,B). The peroxide value did not show a pronounced increase for either sample, especially when compared to the sample without bran (control). The results for the peroxide value were confirmed by monitoring the ratio of unsaturated over saturated fatty acids over time, as no degradation of unsaturated fatty acids was observed (Figure 4C). The slight decrease in TAG content for the toasted wheat bran sample could, thus, not be explained by lipid oxidation, and no solid explanation was found for this decrease. The very limited lipid oxidation in both bran-containing samples shows that wheat bran also protects lipids from oxidation, in addition to RP. This observation is in line with the observations of Rohfritsch et al. [32]. Moreover, as outlined in the study of Abeyrathne et al. [33], monitoring lipid oxidation serves as an indirect mode of measurement of the antioxidant capacity. We can, therefore, conclude that the antioxidant capacity of wheat bran is sufficient to inhibit lipid oxidation in the investigated systems. As outlined in our previous study [34], wheat bran antioxidants such as polyphenols and tocopherols are responsible for RP stabilisation; this can either be by the direct protection of RP from oxidation or by the protection of lipids from oxidation. The latter indirectly results in less RP oxidation, as lipid oxidation and RP oxidation are related processes.

In addition, the limited lipid oxidation in both samples implies no clear difference in the degree of lipid oxidation between native and toasted wheat bran. Consequently, no pro-oxidative effect of free fatty acids on lipid oxidation was observed in this study. This implies that the hypothesis that the lower RP-stabilising potential of native compared to toasted wheat bran could be explained by the extensive lipid oxidation caused by the pro-oxidative effect of the free fatty acids produced by wheat bran lipase does not hold. In the literature, contradictory results regarding the pro-oxidative effect of free fatty acids are often observed, likely due to the use of different systems with different degrees of complexity in the research [12,13,14,35]. As stated by Paradiso et al. [36], the role of free fatty acids in lipid oxidation is complex and not yet fully elucidated. Whether or not a pro-oxidative effect is observed depends on several factors, as different mechanisms are in play. A possible oxidation-promoting mechanism could be the enhanced oxygen diffusion due to a higher concentration of free fatty acids in the oil phase. Roppongi et al. [37] showed that oxygen has a higher solubility in a medium consisting of free fatty acids than in one consisting of triacylglycerols. As oxygen is a crucial reactant in lipid oxidation, a higher oxygen concentration in the oil phase facilitates lipid oxidation. Assuming the oxygen’s solubility in the oil phase is altered in the investigated systems due to lipase activity, this mechanism could be a possible explanation for the faster RP breakdown in native wheat bran systems than in toasted wheat bran systems. Improved oxygen transfer might lead to accelerated RP degradation without promoting lipid oxidation, as wheat bran antioxidants can prevent lipid oxidation.

Given that RP is more susceptible to oxidation than lipids due to its highly unsaturated hydrocarbon chain, wheat bran antioxidants such as polyphenols and tocopherols can only prevent RP degradation to a certain extent [34], whereas they can fully inhibit lipid oxidation in the investigated systems (Figure 4). The absence of lipid oxidation in native wheat bran systems not only shows that wheat bran lipase does not promote lipid oxidation in the investigated systems. It also shows that other wheat bran endogenous enzymes, such as lipoxygenases, which cause enzymatic lipid oxidation, and phytases, which lead to the release of pro-oxidative minerals, do not promote lipid oxidation under the storage conditions used. However, for phytase, the release of pro-oxidative minerals might accelerate RP degradation without accelerating lipid oxidation, as bran antioxidants can prevent lipid oxidation.

As discussed above, extensive lipid oxidation is related to extensive RP degradation. This relationship was clearly observed for the samples without bran. However, lipid oxidation could not explain the differences in RP stability between the samples with native and toasted wheat bran. Therefore, it is assumed that enzymes like lipases and phytases can still enhance RP breakdown without promoting lipid oxidation, as bran antioxidants can prevent lipid oxidation. Although the effects of phytases and the effects of free fatty acids on RP as such pose interesting opportunities for future research, these effects are not investigated in depth in this study. Concerning lipase, it is also hypothesised that the hydrolysis of RP to retinol by wheat bran lipases could be responsible for the difference in RP stabilisation between native and toasted wheat bran. This is further discussed in the next section.

### 3.3. Direct Action of Lipase on Retinyl Palmitate

A storage experiment with systems composed of RTD, soy oil and native or toasted wheat bran was performed to investigate whether wheat bran lipase can cleave the ester bond of retinyl esters. RTD was used instead of RP for this purpose, as tridecanoic acid is not inherently present in soy oil or wheat bran lipids. Palmitic acid, on the other hand, is present in both soy oil and wheat bran lipids. In Figure 5, the percentage of RTD hydrolysis measured as the free tridecanoic acid content over the initial RTD content, both expressed in moles per g lipids, is shown as a function of storage time for systems with native and toasted wheat bran stored for one week.

No RTD hydrolysis was observed for the toasted wheat bran sample, whereas RTD hydrolysis was noticed in the native wheat bran sample. After three days of accelerated storage, 16.3 ± 1.3% of the initially present RTD was hydrolysed into tridecanoic acid and retinol. This percentage increased further to 22 ± 2% after five days of storage and to 23.9 ± 1.3% after seven days. Assuming lipase-catalysed hydrolysis occurs to the same extent for RP and RTD, it can be concluded that wheat bran lipase hydrolyses RP to RE and palmitic acid. This aligns with the research of Turner et al. [17], in which lipase-catalysed hydrolysis of retinyl esters was observed.

As shown in Figure 5, the RTD hydrolysis occurred relatively fast at the beginning of the storage period, and then gradually slowed down. The very limited increase in RTD hydrolysis between five and seven days of storage suggests that RTD hydrolysis evolves to a plateau value. This trend is similar to the trend observed for the free fatty acid production as a function of storage time (Figure 3). The similarity between both measurements is not surprising, as both hydrolysis reactions are due to lipase activity. For the hydrolysis of TAGs to FFAs, the same trend was also observed by Balduyck et al. [38], who suggested that the stabilisation of the FFA content at a plateau value can be explained by the equilibrium of the enzymatic reaction that is reached. At a certain time point, the high free fatty acid concentration can favour the reverse reaction, this being the re-esterification of fatty acids with alcohols. This presumably results in a dynamic balance between hydrolysis and re-esterification. Following this reasoning, RTD hydrolysis may be underestimated, as tridecanoic acid released from RTD might be re-esterified to alcohols, such as glycerol, present in the matrix. However, the RTD content (1.91 ± 0.01 mg/g sample) is very small compared to the lipid content (233 ± 3 mg/g). Consequently, the free tridecanoic acid content is presumably very small compared to the free fatty acid content. Therefore, we expect this effect to be of minor importance. In addition, it is hypothesised that prolonged storage at 60 °C, which is above the temperature optimum of wheat bran lipases, might lead to enzyme denaturation and, thus, a decrease in lipase activity over storage time [39].

Although not all RTD is converted to tridecanoic acid and retinol, 23.9 ± 1.3% hydrolysis after seven days is considerable, and can substantially contribute to vitamin A degradation. This is, however, under the assumption that the released retinol is rapidly degraded. To verify this assumption, the stability of RE and RP dissolved in oil, mixed with native or toasted wheat bran, was monitored during storage. A sample without bran was also included. Given that RTD and RP are both retinyl esters, their stability is assumed to be comparable. As shown in Figure 6, RE was degraded faster than RP for all investigated sample types.

When looking at the retention of RE or RP dissolved in soy oil after three days of storage, only 3.6 ± 1.1% RE was retained, whereas about 52 ± 3% RP was retained. Similar trends were observed for the bran-containing samples. The sample with toasted wheat bran showed an RP retention of 79 ± 2% after three days, but a considerably lower RE retention of 58.2 ± 1.3%. Remarkably, the difference in stability between RE and RP was the largest for the samples with native wheat bran. For RP, a retention of 61.5 ± 1.5% was measured after three days, whereas the retention was only 3.13 ± 0.07% for RE. The results shown in Figure 6 confirm that the stability of RE in the investigated systems is lower than that of RP, and presumably also of other retinyl esters. This difference in stability between RP and RE is in accordance with existing literature [4]. Surprisingly, RE stability was lower for the sample with native wheat bran than for RE dissolved in oil, indicating that native wheat bran cannot stabilise RE in the used system composition. Given the fast degradation of RE in native wheat bran systems, it can be stated that the RE originating from lipase-catalysed retinyl ester hydrolysis is rapidly degraded.

When looking at the RP retention for the native and toasted wheat bran systems (Figure 6), the difference in RP retention after one week of accelerated storage between these two systems was 36 ± 2%. Based on the results shown in Figure 5 and assuming that the stability of RTD and RP is similar, 23.9 ± 1.3 percentage points or 66 ± 5% of this difference can be attributed to the lipase-catalysed hydrolysis of RP to RE and subsequent retinol degradation. RP hydrolysis can thus in part, but not fully, explain the lower degree of RP stabilisation by native wheat bran. The remaining 12 ± 2 percentage points or 34 ± 6% of the difference between both sample types is likely due to the pro-oxidative effect of other enzymes on RP. It is hypothesised that the increase in free minerals by phytase action and the production of FFAs by lipase can accelerate RP degradation. Hereby, the latter hypothesis is based on the study of Roppongi et al. [37], in which it was stated that oxygen solubility is higher in a medium consisting of FFAs than in one consisting of TAGs.

## 4. Conclusions

In this study, the adverse effect of wheat bran endogenous lipase on the stability of RP during the storage of a system composed of RP, soy oil and wheat bran was investigated in depth. It was shown that wheat bran protected vitamin A, in the form of RP, during storage, with toasted wheat bran having a superior protective effect compared to native wheat bran. The stabilisation of RP by wheat bran was, in part, attributed to the prevention of RP and lipid oxidation by wheat bran antioxidants. Moreover, extensive lipid oxidation was shown to be related to rapid RP degradation. However, the discrepancy between native and toasted wheat bran could not be assigned to a difference in the degree of lipid oxidation caused by wheat bran endogenous enzymes. Although a clear free fatty acid production was noticed for the system with native wheat bran, this did not result in an increased lipid oxidation. Next to the hydrolysis of TAGs, wheat bran lipase was also shown to cleave the ester bond of retinyl esters, resulting in the release of the less stable retinol. The occurrence and relevance of this lipase-catalysed hydrolysis reaction was demonstrated using in-house-produced RTD. The direct action of lipase on retinyl esters could explain a major part of the difference in RP retention between native and toasted wheat bran. Next to enzymatic RP hydrolysis, the production of free fatty acids and the release of pro-oxidative minerals by, respectively, lipase and phytase action are assumed to contribute to RP degradation, but only to a limited extent. As enzymes are widely present in raw materials and during food processing, the insights generated in this study are highly relevant to the food industry, and can aid in improving the stability of vitamin A in fortified food products. This is particularly important for RP-fortified cereal products given the high susceptibility of retinyl esters to enzymatic degradation by cereal lipases. To extrapolate the insights gained in this study to RP-fortified food products, future research on the effect of cereal endogenous enzymes as well as the addition of native and toasted wheat bran on the vitamin A stability in food products is needed.

## Figures and Tables

**Figure 1 foods-13-00080-f001:**
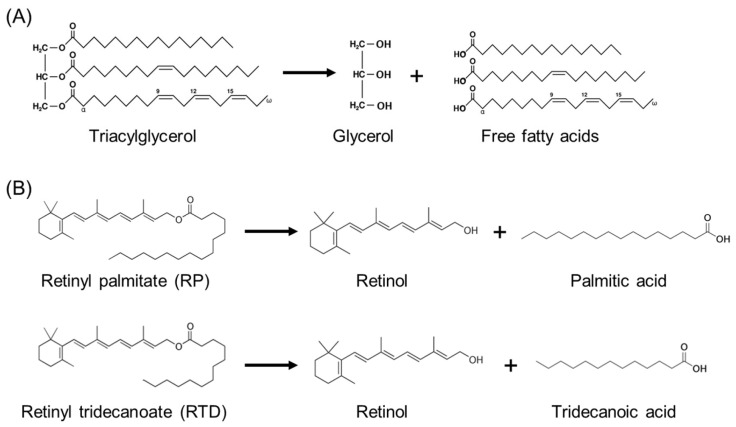
Overview of the lipase-catalysed reactions investigated in this study. (**A**) Hydrolysis of triacylglycerols to free fatty acids and glycerol. This reaction is studied in Section 3.2 and corresponds to the indirect mechanism. (**B**) Hydrolysis of retinyl esters, more specifically retinyl palmitate (RP) and retinyl tridecanoate (RTD), to retinol and, respectively, palmitic acid and tridecanoic acid. This reaction is studied in Section 3.3 and corresponds to the direct mechanism.

**Figure 2 foods-13-00080-f002:**
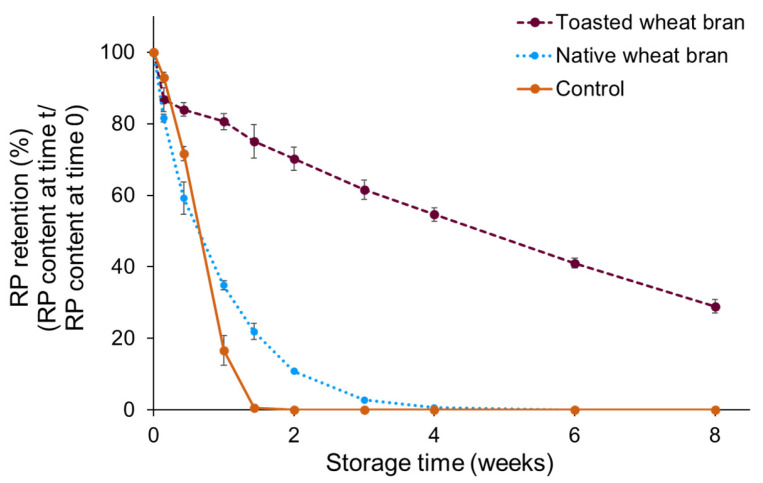
The retinyl palmitate (RP) retention (%) monitored during 8 weeks of accelerated storage (70% relative humidity, 60 °C) in systems composed of 0.16% RP, 19.84% soy oil and 80% native (dotted line) or toasted wheat bran (dashed line), expressed as a percentage of the RP content measured before storage. The control sample (0.8% RP in soy oil) did not contain bran (solid line). The storage experiment was performed in duplicate, and the RP content in each sample was measured in duplicate. Errors bars show standard deviations of the duplicated storage experiments.

**Figure 3 foods-13-00080-f003:**
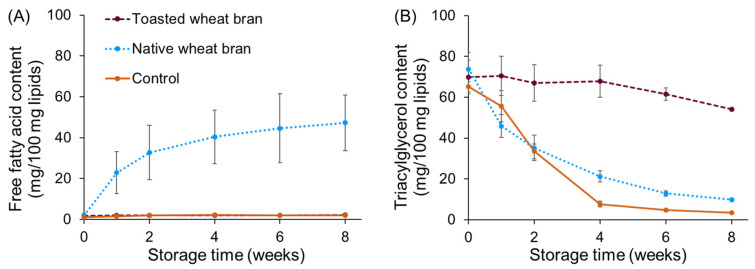
The free fatty acid content (mg/100 mg lipids) (**A**) and triacylglycerol content (mg/100 mg lipids) (**B**) of lipids extracted from systems composed of 0.16% RP, 19.84% soy oil and 80% native (dotted line) or toasted wheat bran (dashed line), and soy oil enriched with 0.8% RP (solid line, control), monitored during 8 weeks of accelerated storage (70% relative humidity, 60 °C). The storage experiment was performed in duplicate, and analytical measurements on each sample were performed in duplicate. Error bars show the standard deviations of duplicate storage experiments.

**Figure 4 foods-13-00080-f004:**
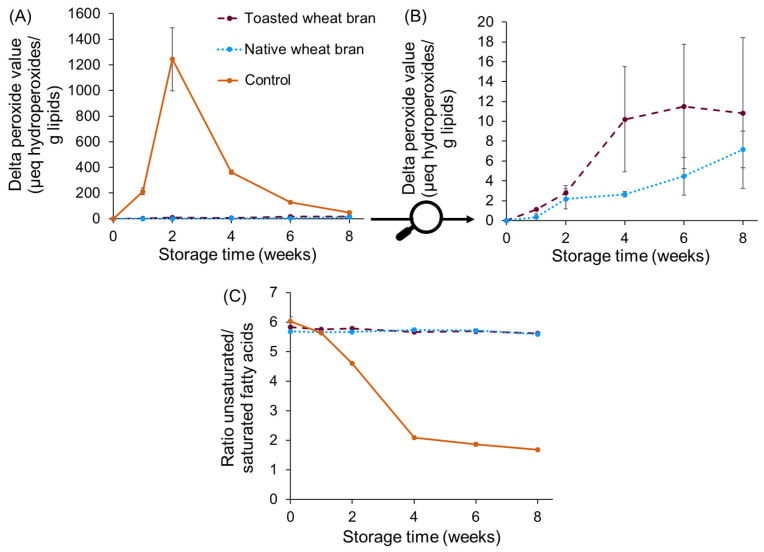
The delta peroxide value (μeq hydroperoxides/g lipids) (**A**,**B**) and ratio of unsaturated over saturated fatty acids (**C**) of lipids extracted from systems composed of 0.16% RP, 19.84% soy oil and 80% native (dotted line) or toasted wheat bran (dashed line), and soy oil enriched with 0.8% RP (solid line, control), monitored during 8 weeks of accelerated storage (70% relative humidity, 60 °C). The storage experiment was performed in duplicate and analytical measurements on each sample were performed in duplicate. Error bars show the standard deviations of duplicate storage experiments.

**Figure 5 foods-13-00080-f005:**
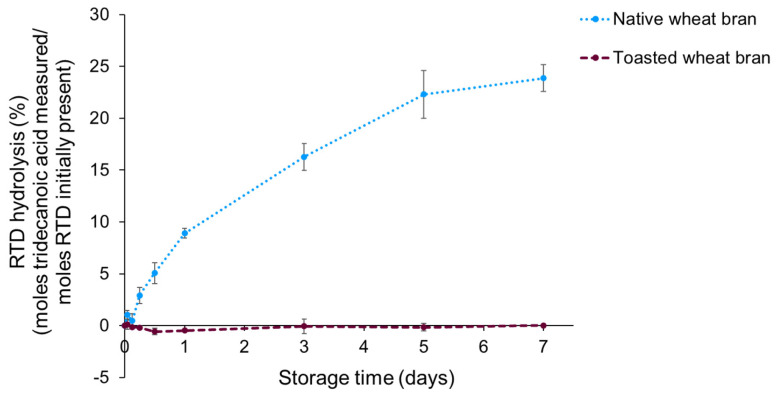
The hydrolysis of retinyl tridecanaote (RTD) to retinol and tridecanoic acid monitored during one week of accelerated storage (70% relative humidity, 60 °C) in systems composed of 0.15% RTD, 19.85% soy oil and 80% native or toasted wheat bran. The hydrolysis of RTD was calculated as the ratio of the molar concentration of the tridecanoic acid measured (moles/g lipids) over the molar concentration of RTD initially present in the sample (moles/g lipids). Hereby, both the measured tridecanoic acid concentration and the initial RTD concentration were corrected for the free tridecanoic acid present at the start of the storage experiment. The storage experiment was performed in duplicate, and analytical measurements on each sample were performed in duplicate. The error bars show the standard deviations of the duplicate storage experiments.

**Figure 6 foods-13-00080-f006:**
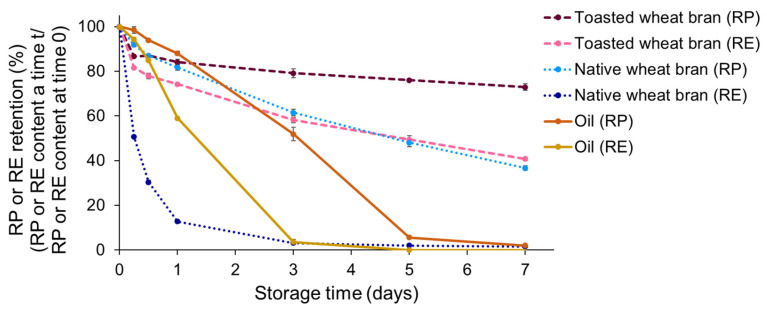
The retinyl palmitate (RP) or retinol (RE) retention (%) monitored during one week of accelerated storage (70% relative humidity, 60 °C) in systems composed of 0.16% RP, 19.84% soy oil and 80% native or toasted wheat bran, or 0.09% RE, 19.91% soy oil and 80% native or toasted wheat bran. Control samples consisting of 0.80% RP or 0.44% RE dissolved in soy oil were also included. The RP or RE retention is expressed as a percentage of the RP or RE content measured before storage. The storage experiment was performed in duplicate and analytical measurements on each sample were performed in duplicate. Error bars show standard deviations of duplicate storage experiments.

**Table 1 foods-13-00080-t001:** Sample composition of the samples used in the accelerated storage experiments. Samples 1, 2 and 3 were used in the first storage experiment, and all eight samples were used in the second storage experiment. Samples 1 and 4 consisted of soy oil mixed with retinyl palmitate (RP) or retinol (RE). All other samples consisted of RP, RE or retinyl tridecanoate (RTD) mixed with soy oil and toasted or native wheat bran. The molar concentration of RP, RE and RTD in soy oil was kept constant.

Sample	Wheat Bran (%)	Soy Oil (%)	RP (%)	RE (%)	RTD (%)
1	0	99.20	0.80	0	0
2	80.00 (native)	19.84	0.16	0	0
3	80.00 (toasted)	19.84	0.16	0	0
4	0	99.56	0	0.44	0
5	80.00 (native)	19.91	0	0.09	0
6	80.00 (toasted)	19.91	0	0.09	0
7	80.00 (native)	19.85	0	0	0.15
8	80.00 (toasted)	19.85	0	0	0.15

## Data Availability

Data is contained within the article or Appendix A.

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
