# Peer review of "Conversion of Retinyl Palmitate to Retinol by Wheat Bran Endogenous Lipase Reduces Vitamin A Stability"

_foods, 2023, doi:10.3390/foods13010080_

Round 1

Reviewer 1 Report

Comments and Suggestions for Authors

Why author chose thses samples, RP-enriched soy oil (0.80% RP), a model system consisting of RP (0.16%), soy oil (19.84%) and native wheat bran (80.00%), and a model system consisting of RP (0.16%), soy oil (19.84%) and toasted wheat bran (80.00%). Does it practical in the real food application? what are these model based on?

Comments on the Quality of English Language

The language is fine.

Author Response

Some additional information was added in the introduction section to provide sufficient background. Additional clarification was given regarding the research design in response to a comment of the reviewer. Figures are now shown in color to present the results more clearly. For a point-by-point response to the remarks, please see the attachment.

Reviewer 2 Report

Comments and Suggestions for Authors

The manuscript was to investigate the effect of enzymatic activity such as wheat bran endogenous lipase on RP stability and effect on vitamin A. The topic is important and it is about the food shelf-life. Overall, the experiment design is reasonable. However, the authors should address the points below to revise the manuscript.

1. In the introduction, please clearly indicate your research objectives.

2. In the methods, the authors discuss on Lipid oxidation such as primary oxidation and peroxide value in system. Can you provide more experiments to discuss other oxidation products formation such as hexanal oxidation?

3. Did you consider the effects of native and toasted wheat bran in other systems? Thus, it can help increase the novelty. 

Comments on the Quality of English Language

The English language should be checked again.

Author Response

Additional background was added in the introduction section. Some clarification was added in the materials and methods section. For a point-by-point response to the remarks, please see the attachment.

Reviewer 3 Report

Comments and Suggestions for Authors

The study by Van Wayenbergh, Blockx, Langenaeken, Foubert and Courtin entitled »Conversion of retinyl palmitate to retinol by wheat bran endogenous lipase reduces vitamin A stability« essentially deals with the use of wheat bran as a vitamin A stabiliser.

The authors found, among other things, that wheat bran stabilises vitamin A (in the form of retinyl palmitate) and that toasted wheat bran is much more efficient at stabilising retinyl palmitate (and retinol) than native wheat bran. According to the study, the most important reason for this stabilising effect is the presence of antioxidants in the wheat bran, and these antioxidants should prevent the oxidation of retinyl palmitate and lipids.

The paper is written according to the usual standards for scientific papers and the English language is also good.

My main comment is really just my observation that the chemical part could perhaps be presented a little more clearly. Therefore, I suggest inserting some pictures showing typical products (chemical structures) of the reactions mentioned in the paper, as it is difficult to read such a paper with understanding without knowing what chemical reactions can take place in the given cases.

Minor remarks: In reference 20, the title of the journal is not written in the abbreviated form.

Author Response

Additional background and references were added in the introduction section. Figures are now shown in color to present the results more clearly. For a point-by-point response to the remarks, please see the attachment.

Reviewer 4 Report

Comments and Suggestions for Authors

The manuscript (foods-2720105) by Van Wayenbergh is interesting and the results obtained may have a practical/commersial applications. Well-written and logically written Introduction. I also have no doubt that the amount of work of the authors in conducting these extensive studies was great, which deserves recognition. However, some doubts need to be clarified.

Main doubts:

In general, you hypothesized that the large difference in RP stabilization between native and toasted wheat bran can be explained by the difference in wheat bran endogenous enzymatic activity. However, this statement (line 296-304) is not supported by evidence. Analysis of enzymatic activity of native and toasted wheat bran endogenous enzymes like lipases, lipoxygenases and phytases is necessary to support the obtained results.

- line 110-111: “Peroxidase inactivation was verified according to Bergmeyer (1974)” – no results of this verification

- lines 392-3: “… the antioxidant capacity of wheat bran might be sufficient to inhibit lipid oxidation” it was not analyzed in the present study, speculative sentence that needs at least the literature-based evidence support,

 404-6: ”… it was suggested that enzymes like lipases and phytases can still promote RP degradation without promoting lipid oxidation, as bran antioxidants can prevent lipid oxidation” – the bran antioxidants were not analysed in this study, provide more details what antioxidants and support this statement with suitable reference

- conclusions: "the adverse effect of wheat bran endogenous enzymes on the stability of RP during storage of a model system consisting of RP, soy oil and wheat bran was investigated in depth" - not entirely, I have a feeling that the presented results show rather the influence of thermal treatment rather than the effect of wheat bran endogenous enzymes on the stability of RP during storage

Additional comments:

- line 93-94) the approximate composition of wheat bran (indicated in Table S1) was not analyzed by the authors, but was provided by the supplier – add this explanation in the text

-was the fatty acid profile analysed solely in the soy oil? does wheat bran affect the fatty acids profiles of the remaining samples? This will consequently result the ration of unsaturated/saturate fatty acids

- thermal treatment of wheat bran (in the oven in 170 °C) for 30 min is a relatively long time, this may be perceived rather as roasting - that is the Authors opinion?

- line 124 and Figures 1-3: could the sample RP-enriched soy oil (0.80% RP) be called control?

-besides, it would be more clear to present results ( in Fig. 1-3) using different colours instead different line types

- line 60 and in the whole text: “…by Turner et al. (2001) that some lipases can hydrolyse the ester bond of RP [17]  – check the suggested by journal method of referencing

Author Response

Additional clarification on the research design was added. The figures are now shown in color to present the results more clearly. For a point-by-point response to the remarks, please see the attachment.

Round 2

Reviewer 1 Report

Comments and Suggestions for Authors

The manuscript is fine at present.

Reviewer 4 Report

Comments and Suggestions for Authors

The authors improved the manuscript according to the reviewers' suggestions. All doubts were dispelled. I recommend this manuscript for publication in its current form